# Silicon Supply Improves Leaf Gas Exchange, Antioxidant Defense System and Growth in *Saccharum officinarum* Responsive to Water Limitation

**DOI:** 10.3390/plants9081032

**Published:** 2020-08-14

**Authors:** Krishan K. Verma, Muhammad Anas, Zhongliang Chen, Vishnu D. Rajput, Mukesh Kumar Malviya, Chhedi Lal Verma, Rajesh Kumar Singh, Pratiksha Singh, Xiu-Peng Song, Yang-Rui Li

**Affiliations:** 1Key Laboratory of Sugarcane Biotechnology and Genetic Improvement (Guangxi), Ministry of Agriculture and Rural Affairs/Guangxi Key Laboratory of Sugarcane Genetic Improvement/Sugarcane Research Institute, Guangxi Academy of Agricultural Sciences, Nanning 530007, China; drvermakishan@gmail.com (K.K.V.); anas.uaf@gmail.com (M.A.); czl_good2007@gxaas.net (Z.C.); malviyamm1983@gmail.com (M.K.M.); rajeshsingh999@gmail.com (R.K.S.); singh.pratiksha23@gmail.com (P.S.); 2College of Agriculture, Guangxi University, Nanning 530004, China; 3Academy of Biology and Biotechnology, Southern Federal University, 344006 Rostov-on-Don, Russia; rajput.vishnu@gmail.com; 4Irrigation and Drainage Engineering, ICAR-Central Soil Salinity Research Institute, Regional Research Station, Lucknow 226005, India; lalcverma62@gmail.com

**Keywords:** antioxidants, biomass, limited water, photosynthetic capacity, silicon, *Saccharum* spp.

## Abstract

Silicon (Si) is not categorized as a biologically essential element for plants, yet a great number of scientific reports have shown its significant effects in various crop plants and environmental variables. Plant Si plays biologically active role in plant life cycle, and the significant impact depends on its bioaccumulation in plant tissues or parts. In particular, it has been investigated for its involvement in limited irrigation management. Therefore, this experiment was conducted to examine the effect of Si application in eco-physiological, enzymatic and non-enzymatic activities of sugarcane plants against water stress. Four irrigation levels, i.e., normal (100–95% of soil moisture), 80–75, 55–50, and 35–30% of soil moisture were treated for the sugarcane cultivar GT 42 plants supplied with 0, 100, 200, 300, 400 and 500 mg Si L^−1^ and exposed for 60 days after Si application. Under stress, reduction in plant length (~26–67%), leaf area-expansion (~7–51%), relative water content (~18–57%), leaf greenness (~12–35%), photosynthetic pigments (~12–67%), physiological responses such as photosynthesis (22–63%), stomatal conductance (~25–61%), and transpiration rate (~32–63%), and biomass production were observed in the plants without Si application. The drought condition also inhibited the activities of antioxidant enzymes like catalase (~10–52%), peroxidase (ca. 4–35), superoxide dismutase (10–44%) and enhanced proline (~73–410%), and malondialdehyde content (ca. 15–158%), respectively. However, addition of Si ameliorated drought induced damage in sugarcane plants. The findings suggest that the active involvement of Si in sugarcane responsive to water stress ranges from plant performance and physiological processes, to antioxidant defense systems.

## 1. Introduction

As a result of rapid worldwide economic growth, limited irrigation is becoming an increasing, major environmental problem for plant growth and crop production. Plant water deficiency may result in scarcity of water supply in soil (water stress) or may create an interference to water uptake in plants [1,2]. The large areas of farming land are reported to suffer from seasonal water stress [3]. The global population has reached about 7.7 billion presently, which may further swell by up to ca. 2 billion in the coming 30 years, reaching approximately 10 billion by 2050 [4,5,6]. To feed such a large number of people would be a major challenge for agri-sectors, farmers and scientists against the dynamic era of climate change in near future [7]. The population ballooning may severely impair the land holding capacity, particularly in Asian countries [5]. The soil health may also be affected due to the enhanced use of the chemical fertilizers, insecticides and other contaminants such as plastics, arsenic, fluoride, salinity, GHGs and global warming [5], which would impair crop productivity, which is associated with the soil condition and other related environmental variables [8].

Agricultural land gets affected by salinity up to 1–2% [9], while water stress accounts ca. 30% of the global land. Both stresses share many common similarities in terms of their impact on crop production [10]. The water stress in plants includes damage to the cell membrane functions and negative effects on photosynthetic capacity and antioxidative enzyme defense mechanisms [2,11,12]. Variations were noted in germination (%) and root development, such as diameter, area expansion, total and main root growth. Some morphological changes, such as early leaf senescence, chlorosis, and necrosis, have been observed [5,11,12]. Drought damage can also be caused by the higher vapor pressure deficit in the environment, which results in more water loss via water vapor than the rate of water transport to the plant leaves [13]. In this situation, plant water status is disturbed, resulting in the disruption of metabolic functions and loss in growth and development [14]. Therefore, an essentiality appears to be to maintain and sustain the agriculture crop production against limited water availability, to continue to feed the growing population adequately.

Sugarcane (*Saccharum officinarum* L.) is a C_4_ crop mainly cultivated in arid and semi-arid areas throughout the world. The cultivation of *Saccharum* has been continuously enhanced and is considered as a renewable feedstock for valuable products such as sugar and bioethanol, and renewable energy sources to replace fossil fuels [3,15,16]. However, the remaining biomass after sugar and bioethanol production can be burned to generate renewable energy and/or used for 2G-bioethanol production [17,18].

Silicon (Si) constitutes a crucial part of the soil in the form of silicate or aluminum silicates. Silicon serves as a biologically active and significant element for agriculture and is listed as 8th most common element in nature and the second most common element found in soil after molecular oxygen [19,20]. It is required for the growth of diatoms, sponge, and corals, and is also found to be associated with plant cell growth because Si enhances biotic resistance against bacteria, fungi, viruses, and herbivores [21,22,23,24]. Silicon is considered as quasi-essential [25] for plant development. Silicon is a high-quality fertilizer for developing economically sustainable agriculture. The impact of Si on plant growth, development and production has been well documented [26]. Silicon has been testified to gradually promote the development of biomass, productivity and quality of a wide latitude of crops such as monocots, dicots and some vegetables and fruit crops, which actively take up and accumulate excessive amount of Si by the plant parts [26]. Thus, Si exerts beneficial effects on plants’ fitness and productivity by reducing environmental stresses [2,11,12,27,28] along with regulation of defense signaling pathways.

However, knowledge about how Si modulates the morphological, physiological and biomass accumulation in *Saccharum officinarum* “GT 42” during water stress remains elusive. Although the essentiality of this element to plants is still debated, there have been significant impacts in our understanding of the uptake of Si in plants. In addition, present data regarding the precise amount of Si for its application method in *Saccharum officinarum* plants are limited. This study has examined the roles of Si on growth, photosynthetic capacity and antioxidative enzymes of sugarcane plants under combined irrigation with Si and limited water supply, in order to provide evidence demonstrating the biological and atmospheric responses of Si in increasing tolerance to abiotic stresses.

## 2. Materials and Methods

### 2.1. Experimental Site and Treatment Design

A pot experiment was conducted during the period March–September 2019 in a greenhouse at the Sugarcane Research Institute, Guangxi Academy of Agricultural Sciences, Nanning, Guangxi, China (23.6° N 108.3° E). Sugarcane is widely cultivated in southern China, such as in the Guangxi, Yunnan, Guangdong and Hainan provinces. Sugarcane (*Saccharum* spp. cv. GT 42) culms were germinated inside the greenhouse in the field plot. Following standard agronomic practices, bud cane setts were planted in the month of March. Recommended row to row (75 cm) spacing was maintained. Basal nitrogen, phosphorus and potassium (NPK) doses were applied at the time of sowing. Before sowing, cane buds were dipped in water for up to 48 h and treated with Bavistin solution (fungicide) for 5 min. Sixty-days after sowing, the plants were shifted into the plastic pots, filled with 3.5 kg soil pot^−1^ (agricultural fertile soil and compost, 2:1, *w/w*) and kept in a greenhouse. The availability of macro- and micro-nutrients in the filled soil was analyzed prior to the experiment (Appendix A and Appendix A).

All the plants were irrigated every day up to soil moisture capacity to establish normal plant growth. The four levels of soil moisture were maintained as 100–95, 80–75, 55–50 and 35–30% of field capacity up to sixty-days under limited water supply (Figure 1D). The soil moisture content was observed according to Verma et al. [2]. During the drying cycle, tillers were cut immediately after emergence. The experiment consisted of six silicon solutions with different Si levels such as 0 (control), 100, 200, 300, 400 and 500 mgl^−1^. The Si solution was prepared by dissolving the appropriate quantities of CaO·SiO_2_, and applied twice with the irrigation at an interval of four weeks. This experiment was designed to be completely randomized with ten biological replicates.

During the experiment, minimal day-to-day changes in climatic variables such as diurnal average air temperature, air relative humidity and average light exposure were monitored (Figure 1A–C). Soil moisture (%) in the Si applied plants was maintained to be equal to control plants. Soil moisture (%) at 0–10 cm from soil depth was observed by Soil Moisture Meter at three points in each pot during the experiment (Figure 1D).

### 2.2. Determination of Relative Water Content, Chlorophyll and Chlorophyll Stability Index

Fresh, fully expanded top visible dewlap (TVD, leaf +1) leaves from randomly selected plants were used to determine the relative water content of leaf and photosynthetic pigments. The collected leaves were sealed in plastic bags and carried to the laboratory. After fresh weight measurement, leaves were immersed in tap water overnight in a refrigerator, placed over tissue paper for drying, and then weighted. Leaf samples were dried at 65 °C in an oven, after which the dry mass was observed [29].

Photosynthetic pigments were estimated by dimethyl sulphoxide (DMSO) as the extraction reagent [30]. The absorbances at 663 and 645 nm were recorded by spectrophotometer, with DMSO being used as a blank. The relative leaf greenness was estimated using a Chlorophyll Meter (SPAD-502, Minolta, Inc. Japan) on the same leaves as the photosynthetic measurements with ten measurements per leaf per plant. The chlorophyll stability index (CSI) was assessed as per the method of Sairam et al. [31].

### 2.3. Photosynthetic Characteristics

Photosynthetic traits such as net photosynthetic CO_2_ assimilation (*A*), transpiration rate (*E*) and stomatal conductance (*g*s) were observed 60 days after limited water irrigation and the silicon supplementation period, using an LI-6800 portable photosynthesis system (LI-COR Biosciences, Lincoln, NE, United States). For each irrigation treatment, leaf gas exchange was conducted between 09:00–11:00 a.m. on treated and normal plants (five replicates). In each pot, the leaves (middle third of leaf + 1) were used for photosynthetic parameters. Inside the leaf chamber, photosynthetic photon flux density (PPFD) was set at 1000 µmol m^−2^ s^−1^, leaf temperature (25 °C), and CO_2_ concentration (400 µmol mol^−1^).

### 2.4. Measurement of EnzymeActivities

Approximately 500 mg of fresh sugarcane leaves were frozen in liquid nitrogen and extracted in 50 mM potassium phosphate buffer (PBS, pH 7.8). Following the centrifugation (15,000× *g* for 10 min at 4 °C), the supernatant was collected for enzyme assays.

The catalase (CAT, EC 1.11.1.6) activity was determined according to the method as stated by Azevedo et al. [32]. Peroxidase (POD, EC 1.11.1.7) activity was quantified according to Bai et al. [33] with minor changes. The reaction mixture contained 1.0 mL of 0.3% H_2_O_2_, 1.0 mL of 0.05 M PBS (pH 7.8), 0.9 mL of guaiacol (0.2%), mixed with the 0.1 mL of enzyme extract. The absorbance of the mixture was observed at 470 nm.

The superoxide dismutase (SOD, EC 1.15.1.1) activity was estimated based on the inhibition of nitroblue tetrazolium (NBT) photo inhibition reduction [17,34] with minor modifications. The reaction mixture included 2.2 mL of PBS (50 mM pH 7.8, 0.2 mL of methionine (130 mM), 0.1 mL of EDTA-Na_2_ (20 µM), 0.2 mL of NBT (750 µM), 0.2 mL of riboflavin (100 µM) mixed with 0.1 mL of the enzyme solution. The specific SOD activity was expressed as the amount of enzyme required that produced 50% inhibition of NBT reduction under assay conditions.

### 2.5. Determination of Proline and Malondialdehyde Content

Proline content was estimated by the acid-ninhydrin method according to Bates et al. [35]. The 500 mg fresh leaf samples were homogenized in 10 mL aqueous sulphosalicylic acid (3%) and centrifuged (3500× *g*, 10 min). The extract (2 mL) was treated with two ml acid ninhydrin and two ml glacial acetic acid, the reaction mixture was shifted in water bath (100 °C, 1 h), and immediately cooled to stop the reaction. The mixture was extracted with four ml of toluene. The absorbance of the pink colored layers was measured at 520 nm, using toluene as blank. The proline content was quantified from a standard calibration curve.

The malondialdehyde (MDA) contents in sugarcane leaves were monitored based on the thiobarbituric acid according to the Bailly et al. [36], with minor changes. The 500 mg Leaf samples were homogenized in 8 mL of trichloroacetic acid (0.1%, *w/v*); the mixture was centrifuged at 5000× *g* (10 min, 4 °C), and supernatant was used for further analysis. Equal volume (1:1) of supernatant was mixed with 0.5% (*w/v*) of thiobarbituric acid in 5% (*w/v*) trichloroacetic acid, boiled (20 min), and immediately cooled to stop the reaction. After centrifugation (8000× *g*, 10 min), the absorbance of the mixture was determined at 450, 532 and 600 nm.

### 2.6. Determination of Silicon Content

Silicon content was determined according to method used by Wang et al. [37] with minor changes. The leaf samples were collected, washed and fully dried (65 °C) up to constant weight. Then, 200 mg of dried leaf powder from each sample was digested with a microwave digestion system using seven ml of oxidizing solution (HNO_3_– six ml and 30% H_2_O_2_- one ml) for half an hour (150 °C—10 min and 180 °C—20 min). Prior to analysis, the digested extract was diluted with deionized water up to 100 mL (final volume). The concentration of silicon in each digested sample was quantified by inductively coupled plasma-optical emission spectroscopy (ICP-OES optima 8300, Perkin Elmer, MA, USA), calibrated by standard solution.

### 2.7. Measurement of Growth and Biomass Traits

The plant length and leaf area-expansion were measured by a measuring tape meter and Leaf Area Meter (CI-203 Area Meter, CID, Inc., USA). After sixty-days of treatment, the sugarcane plants were harvested, washed with irrigation water, and weighted. The fresh leaves, stems and roots were placed in paper bags and oven-dried at 65 °C, and the dry biomass of the plant organs was determined until the constant weight.

### 2.8. The Model

Physiological and biochemical activities in sugarcane are dependent on the applied doses of the silicon in the soil. The present experiment shows that silicon application under limited irrigation levels (% of soil moisture) significantly up-regulated the photosynthetic and biochemical activities of sugarcane. It is hypothesized that the frequency of change (% increase) in physiological and antioxidant activities with respect to Si application in soil is directly proportional to the concentration of silicon supplied in the soil. Mathematically, it can be expressed as:(1)$$\frac{dP}{dS}\;\propto S$$
where *P* = Photosynthetic parameters, i.e., photosynthetic CO_2_ assimilation rate, transpiration and stomatal conductance, *S* = concentration of silicon supplied, *dP* = incremental variations in photosynthetic characteristics due to increasing concentrations of silicon (*dS*), *dS* = increasing concentrations of silicon.

### 2.9. Governing Equation

The above hypothesis can be translated into a governing equation by introducing a proportionality constant (*K*) as below.
(2)$$\frac{dP}{dS}=KS$$
where *K* = proportionality constant and is also known as base response (λ). The actual base responses of photosynthetic capacity in control plants. 

### 2.10. Solution

Variables of Equation (2) were separated and written as below.
(3)∫dP =λ ∫S dS

Solving above governing equation following solution would be obtained.
(4)$$P_{S}\;=\frac{\lambda }{2}S^{2}{\;+\;I}_{}$$
where *Ps* = Physiological responses against applied doze of silicon; *I* = integration constant, which can be worked out by substituting the initial conditions in Equation (4), i.e., *S = 0, P_s_ = λ.*
(5)$$\lambda =\frac{\lambda }{2}{\;\times \;0\;+\;I}_{}$$
(6)I = λ

Substituting value of the integration constant in Equation (4), the general solution can be written as below.
(7)$$P_{}\;=\lambda \;+\;\frac{\lambda }{2}S^{2}{}_{}$$
or
(8)$$P_{S}\;=\lambda \;(1\;+\;\frac{1}{2}S^{2}{)}_{}$$

The expression within parenthesis is an approximation of cosine function, hence the solution can be written as below.
(9)$$P_{S}\;=\lambda \;{\left(cosS\right)}_{}$$

The solution can be further generalized by providing flexibility to re-orient the curve against the *x* and *y* axis besides adjusting amplitude and wavelength. Equation (9) can be re-written as below.
(10)$$P_{S}\;=\alpha +\lambda cos{\left(\beta S+\omega \right)}_{}$$

The constant α decides the shifting of the cosine curve with respect to the *x*-axis. The constant *β*, *ω* decides the frequency or wavelength and lateral shifting (*y*-axis) of the cosine function. The coefficient λ (base response) decides the significance of the amplitude of cosine function.

### 2.11. Statistical Analysis

All values were represented as the means. One-way analysis of variance (ANOVA) was used to analyze the significance of the differences among different irrigation levels. Two-way ANOVA was performed to analyze the interaction between Si and limited water irrigation.

## 3. Results

### 3.1. Impact of Silicon on Growth and Biomass under Limited Water Irrigation

Growth and biomass traits were measured in terms of plant length, number of leaf-area expansion, fresh and dry mass reduced significantly (*p* < 0.05) following limited irrigation in sugarcane plants. Irrigation with 80–75, 55–50 and 35–30% of soil moisture decreased plant length (~26–67%), number of leaves-area expansion (~39–67, 7–51%), fresh biomass (~18–59%) and dry biomass (~18–66%) in sugarcane plants, respectively, as compared to the normal irrigation like 100–95% of soil moisture. On the other hand, Si supplied along with limited water irrigation alleviated the reduction in plant growth and biomass, as percentage increased (Table 1 and Table 2). However, the concentration of Si enhanced growth and biomass of sugarcane plants was over the values of normal irrigation. Thus, the data indicated that 400 mg Si L^−1^ was more effective compared to 100–300 and 500 mg Si L^−1^ in reducing the 80–75, 55–50 and 35–30% of soil moisture.

The application of Si as soil irrigation facilitated plant performance and was significantly up-regulated compared to that in limited water supply. The plant development showed a substantial improvement with the increasing level of Si application. As shown in Table 2, the fresh and dry mass of leaves, stems and roots of sugarcane plants were markedly declined against limited water irrigation as compared to normal irrigation, while the negative effects of limited water supply on biomass were positively mitigated and gradually enhanced with increasing levels of Si applied as soil irrigation. Biomass was initially up-regulated with the Si levels (100–400 mg Si L^−1^) and then declined considerably at a higher concentration of Si (500 mg Si L^−1^).

### 3.2. Impact of Leaf Gas Exchange Characteristics

Photosynthetic traits declined during the whole plant growth stage during limited water irrigation like 100–95, 80–75, 55–50 and 35–30% of soil moisture (Figure 2A). At all irrigation treatments, *A*, *gs* and *E* under Si application were found higher without Si, at 60 days after Si applied. Photosynthesis for all Si-treated plants were found higher as 2–5, 10–25, 1–48 and 0.5–60% than that without Si application, i.e., 100–95, 80–75, 55–50 and 35–30% of soil moisture (Figure 2A). As shown in Figure 2B,C, limited water irrigation caused a severe loss in *gs* (~61%) and *E* (~63%) in 35–30% of soil moisture in sugarcane plants. However, soil irrigation with various concentrations of Si obviously mitigated the water stress-induced decrease in *g*s and *E*. The 100–95, 80–75, 55–50 and 35–30% of soil moisture irrigations with Si fertilizer up-regulated *g*s by 1–44, 3–49, 3–50 and 11–74% and *E* by 1–29, 0.1–27, 0.7–66 and 9–67%, over the free from Si application, respectively. Fertilization of Si with irrigated water also noted remarkable differences in *A*, *g*s and *E* parameters. However, *A*, *g*s and *E* were markedly up-regulated at 100–400 mg Si L^−1^, and later significantly inhibited at the 500 mg L^−1^ Si concentration. The *A*, gs and *E* values were found the highest at 400 mg Si L^−1^ as compared to 100–300 and 500 mg Si L^−1^ with limited soil moisture. These data showed that Si applied with irrigation could mitigate the negative effects of drought on sugarcane plants.

### 3.3. Photosynthetic Pigments, Leaf Greenness (SPAD Value) and Relative Water Content

At sixty days after soil irrigation with Si, chlorophylls and SPAD values were significantly higher in the Si-supplied plants than in those without Si (Figure 2D,G). Our results showed that limited water irrigation decreased the chlorophyll and SPAD values compared to those of normal irrigation (100–95% of soil moisture). The irrigation treatment with 400 mg Si L^−1^ was more effective, which increased Chl a, b, a + b and SPAD values by 6, 8, 9 and 10%, respectively, as compared to the treatment with 100–300 and 500 mg L^−1^ of silicon. Likewise, the highest content of chlorophyll and SPAD values were also calculated for 100 mg of Si supply.

Chlorophyll stability index (CSI) was observed in sugarcane plants with silicon against limited water irrigation (Figure 2H). The CSI percentage in sugarcane plants were markedly decreased up to 35–30% of soil moisture as compared to well irrigated plants. CSI was gradually increased with increasing levels of Si with limited water supply. The highest percentage was found to be ~11, 13 and 16% in 500 mg Si L^−1^ as compared to 100 mg Si L^−1^, respectively.

Leaf relative water content (RWC) is the main sign that shows water status and survival percentage of the plants against unfavourable environmental conditions. A significant loss in the RWC of stressed sugarcane plants was found as compared to normal irrigation (Table 1). On the other hand, there was a significant maintenance or improvement in RWC in the treatment with Si against limited water irrigation. The most substantial increase of up to 11% was found in 400 mg L^−1^ applied Si, as compared to normal irrigation free from silicon.

### 3.4. Impact of Si on the Activities of Antioxidants against Limited Water Irrigation

In order to assess the role of Si in the regulation of antioxidant enzyme activities and the relationship between reactive oxygen species (ROS) and the cellular defense system, the activities of CAT, POD and SOD were examined. The data in Figure 3A,C,E show that limited water irrigation significantly (*p* < 0.05) decreased CAT (ca. 10–52%), POD (~4–35%) and SOD (~10–44%) activities in sugarcane at 60 days after limited water irrigation.

The enzymatic antioxidant activities were markedly up-regulated by the supplementation of Si in the plants subjected to limited water irrigation. However, the different Si concentrations (100–500 mg Si L^−1^) showed an increasing pattern for improving CAT, POD and SOD enzymes. The antioxidant activities were initially enhanced with increasing levels of Si, and reached the highest percentage values of CAT— 11, 18, 55%; POD-19, 32, 50% and SOD—16, 34, 59% of 80–75, 55–50 and 35–30% as compared to 100–95% of soil moisture with 400 mg Si L^−1^ and limited water irrigation. The higher concentration of Si (500 mg L^−1^) slightly decreased the enzymatic activities as compared to that of 400 mg Si L^−1^, but the antioxidant activities in these irrigations were still higher as compared to the value of normal plants (Figure 3A,C,E).

### 3.5. Impact of Si on Proline and MDA Content

The Figure 3B also indicated a significant increase in proline content in sugarcane leaves during stress. Under stressed condition, the enhancement in proline content was ca. 73, 278 and 410% as compared to control plants. A significant decrease in proline content (ca. 38, 25 and 33% as compared to normal plants) was noticed in drought with Si applied plants, respectively. Although the role of proline has been postulated in various crops during abiotic stresses (i.e., water stress), a down-regulation in proline after application of Si during water stress arguably indicates recovery of relative water content, and loss in proline through degradation or reduced synthesis.

Lipid peroxidation (MDA content) is a marker to observe the impact of limited water supply on lipids. Accordingly, 80–75, 55–50 and 35–30% of soil moisture also increased the MDA content ca. 15, 64 and 156%, respectively, as compared to the value of normal plants (Figure 3D). Supplementation of Si with the limited water irrigation resulted in declined accumulation in the MDA content. On the other hand, the supplementation of Si did not influence the production of MDA in sugarcane plants. The MDA content declined by up to ~20, 18 and 32% for increasing levels of Si with 80–75, 55–50 and 35–30% of soil moisture as compared to normal plants, respectively (Figure 3D).

### 3.6. Impact of Silicon on Si Accumulation under Limited Water Irrigation

The data in Figure 3F showed that, under limited water irrigations of 80–75, 55–50 and 35–30% of soil moisture, the Si accumulation increased up to 265, 135 and 129% higher, respectively, as compared to the control. The accumulated leaf Si concentration was the highest in 500 mg L^−1^ Si with limited water supply in sugarcane plants. The highest uptake (up to 265%) was noted in 80–75% of soil moisture with 500 mg L^−1^ Si concentration (Figure 3F).

### 3.7. Regression Analysis

Regression analysis was analyzed to fit the measured photosynthetic and antioxidant parameters in sugarcane plants with respect to their silicon concentration supplied in the soil to verify the hypothesis and work out the model parameters. Model characteristics (α, β, λ and ω) and coefficient of determination (r) and standard error (S) were worked out from the measured values after regression analysis (Appendix A). The variations of overall growth, photosynthetic and antioxidant parameters explained by the proposed model ranging ‘r’ from 0.178 to 0.999 and ‘S’ from 0.001 to 48.015, verifying the proposed hypothesis and developed model for explaining the changes of the observed parameters against silicon with limited water irrigation levels.

## 4. Discussion

All plants required sufficient amount of essential mineral nutrients to complete life cycle. The growth and biomass of plants display marked changes in response to limited water irrigation [2,38]. Water and mineral elements are absorbed by the plant roots, and limited water supply can have a negative impact on plant fitness through reducing root development [39]. The up-regulation of plant growth and biomass caused by silicon addition has also been noted against limited water supply [2,40,41]. The impact of Si on root growth may be due to enhanced root elongation caused by an increase in cell wall enlargement in the growth area [42]. The augmented water uptake during the supplementation of Si in limited water supply treatment is the result of maintained or improved root hydraulic conductance [43], and root growth [44].

Limited water supply also limits nutrient accumulation through root and subsequent transport to leaves via shoot, thereby minimizing nutrient supply and metabolism [45]. Silicon may play significant role in balancing the uptake, transport and distribution of mineral elements against the limited water supply [46,47]. However, the application of silicon for soil irrigation minimized the severity of limited water supply-induced growth inhibition (Table 1 and Table 2). It increased sugarcane tolerance to limited water irrigation in terms of promoting the plant development. The enhancement in biomass and growth traits are attributed to the up-graded carbon assimilation due to the increased photosynthesis of Si-supplied plants [2].

Results confirmed the previous reports that an amendment of Si alleviates the water stress caused by limited water supply irrigation, and significantly affects plant growth, development and productivity, such as for *Triticum* spp. [48,49], *Oryza sativa* [44], *Glycine max* [50], *Helianthus annuus* [51] and *Zea mays* [52]. Silicon supplementation resulted in maximum growth and biomass, which could likely enhance soil water consumption due to enhanced leaf area-expansion and thus transpiration rate (Table 1 and Table 2), which improved the stress condition [53,54,55]. The Si-mediated improvement in plant development not only takes place under the control condition [56], but also under the limited water supply [57].

The reduction in green pigments may mark the sign of leaf senescence. The reduction in chlorophyll content may be due to the formation of proteolytic enzymes—i.e., chlorophyllase, which is responsible for chlorophyll reduction [58] as well as damage to the photosynthetic machinery [2]. In this article, exogenous amendments with different concentrations of Si were found to significantly increase the leaf greenness (SPAD units) and photosynthetic pigments of sugarcane plants (Figure 2D–H). The supplemented silicon enhanced the photosynthetic capacity of the limited water supply and also linked to the improved efficiency of photosynthetic pigments and photosynthetic enzymes such as ribulose-bisphosphate carboxylase and NADP^+^ dependent glyceroldehyde-3-phosphate dehydrogenase against limited soil moisture content [59]. Chloroplasts are good biosensors of stress because they are the key site for photosynthetic assimilation, a process which is more susceptible to damages caused by abiotic stresses. According to Ruiz-Espinoza et al. [60], the SPAD index depends on various factors, such as the species/variety, the diversity, the thickness, and the age of leaves.

Maintenance of the photosynthetic CO_2_ assimilation rate is crucial to plant growth, development and yield production. Stomatal closure was an early response to limited water supply and an effective way to decrease water loss (Figure 2B). However, it also limits CO_2_ diffusion into the plants, which causes the severe reduction in photosynthesis in sugarcane plants [2,61]. Addition of Si enhanced the silicon content in the leaves and thus could enhance their photosynthetic rate, but only when the plants were subjected to limited water supply (Figure 2A). The significant effects of Si at the end of drying cycle, which may be attributed to more biomass and consumption of sufficient water by transpiration associated with severe stress in relation to the normal condition (Table 1 and Table 2). Furthermore, several scientific reports noted that plants with Si applied led to increased stomatal conductance (Figure 2B), transpiration rate (Figure 2C), and water content of leaf (Table 1), root and whole-plant hydraulic conductance [2,62,63]. Similar findings were found in various crop plants under stressed conditions [64,65,66].

Leaf water level is assessed by water uptake and transport, as well as transpirational loss [50]. The up-regulation in leaf water content and water capacity in the treatment with Si with the limited water supply was caused by the leaves’ thickness as compared to the normal plants without Si [67]. Based on the recent findings, the responses of Si on transpiration rate may be associated to the plant species or varieties and atmospheric circumstances [50]; therefore, after the amendment with certain Si concentrations, an enhanced transpiration rate (Figure 2C; [44]) was observed in some plants, but reduced transpiration in some plants [68], and no change in others [43]. However, the exogenous use of Si was found to improve the photosynthetic capacity in various plants against limited water supply [2,47,48].

An enhanced synthesis of osmolyte compounds, i.e., proline, has been associated with cellular membrane protection [69,70]. Furthermore, proline accumulation has also been related to an improvement of cellular water status and ROS scavenging in sugarcane crop [71,72]. It is widely reported that Si improved and/or maintained the plant tolerance to water deficit by adjusting osmolytes concentration in various crop plants [46,47].

Silicon has been reported to protect the damage of membrane caused by the formation of MDA [73] by regulating antioxidant defense in plants [46]. In present work, it has also been shown to reduce MDA content (Figure 3D), the end-product of lipid peroxidation in stressed plants [50,74,75,76], and thus may help to maintain/balance membrane integrity and reduce membrane permeability [26].

One of the immediate effects on plants subjected to water stress is the production of ROS, i.e., singlet oxygen, superoxide anion, H_2_O_2_ and hydroxyl radicals. The production and accumulation of ROS in the plants result in the severe destruction of the cellular ultrastructure, organelles and functions. The developed complex antioxidant system to balance and/or maintain homeostasis through enzymatic and non-enzymatic antioxidants is one of the strategies of the plants to mitigate and repair the damages caused by ROS [2,3,77,78]. The plants cultivated during stress suffer from water shortages, resulting in the overproduction of ROS in the plants [2,79]. By modulating the plant antioxidant defense systems, Si could also mitigate oxidative damage in plants subjected to water deficit conditions [11,77]. The treatment of water stressed plants with Si causing enhanced activities of CAT, POD and SOD enzymes was observed (Figure 3A,C,E), as in similar findings noticed by various researchers [46,77,78,80]. However, how Si mediates this response is still unclear. Under water stress, Si’s effect on antioxidant enzymes in plants varies not only among plant species/cultivars, but also at various growth phases of the same plant [81]. Abiotic stresses like water deficit can also modify the antioxidant enzyme activities [82]. Overall, based on the presented results and available database, it could be concluded that Si may mitigate the oxidative damage in plants by modulating antioxidant defense system [24,46,77,78].

The correlation coefficient (r) for each set of parameters with silicon was found in the range of 0.178–0.999 (Appendix A). This means that the derived model verified the variations in photosynthetic and biochemical parameters against limited water irrigation levels with silicon almost 99.9%. Model hypothesis explains for rate of variations in their percent increase in the observed parameters upon subjecting sugarcane to limited water supply with or without silicon application. This may be the direct proportion to the escalating silicon concentrations for predicting the gains in other crops in similar situations, which can also be verified.

The results of this study indicated that Si mitigated the detrimental effects of water stress on sugarcane. The addition of Si is a practical approach to maintain and/or increase environmental stresses [77]. Similar reports have shown that Si addition alleviates abiotic stresses, such as water and salt stresses, by up-regulating the activities of antioxidants enzymes [2,73,79,80]. Therefore, the induced enhancements in antioxidants status can be considered as an important mechanism or functions in the cellular defense strategy against abiotic stresses [55,83,84]. Silicon can not only maintain or improve the water deficit-tolerance of high accumulation rate of Si in plants (Figure 3F), but also upgrade the low Si-accumulating plants [50]. In this study, the stress-tolerant mechanism of the sugarcane plant on morpho-physiological and biochemical bases have been elucidated, which is consistent with the study aims.

In this study, it was evident that the uptake of Si was enhanced by increasing the application level of exogenous CaO·SiO_2_, while it was inhibited by decreasing soil moisture, suggesting Si may be involved in the metabolic and physiological capacity in sugarcane plants against water stress. Low soil moisture is one of the serious environmental circumstances limiting the growth and production of crop plants. Predictions of future climate might change to an enhanced severity and duration of stress in the era of climate change. Silicon, as a biologically active element, might be associated with physiological metabolism and/or structural formation when subjected to environmental stresses. The findings suggest that the application of Si to drought-exposed sugarcane crops would lead to better plant development and yield production under stressed conditions. It is therefore clear that part of the beneficial effects of Si on plants is linked to direct/indirect effects on the cell wall. In addition, the amendment of Si and its responses against in vivo conditions still needs extensive assessment. The role and/or functions of silicon in genomics and proteome of the sugarcane plants against limited water supply will be studied further in the future.

## Figures and Tables

**Figure 1 plants-09-01032-f001:**
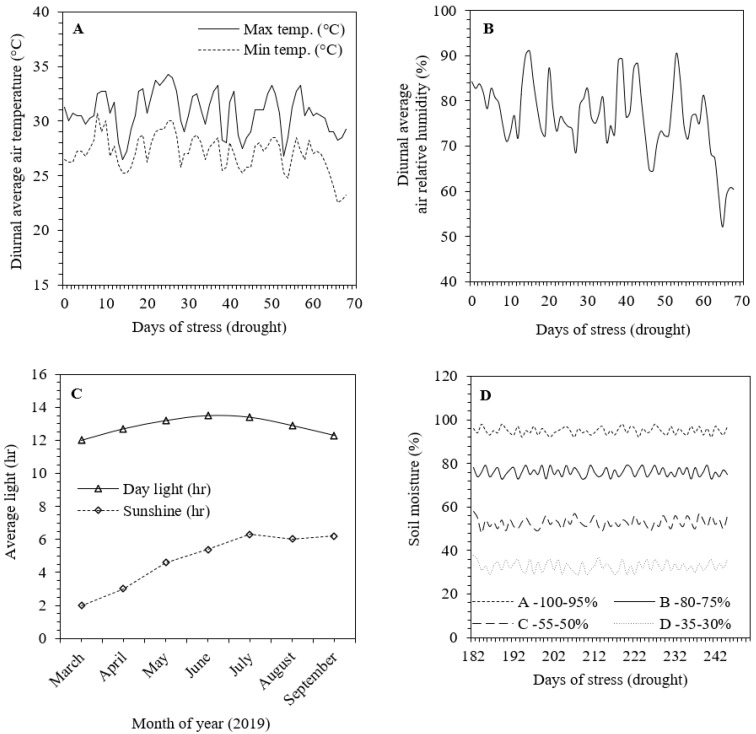
(**A**) Diurnal variation in average temperature (°C), (**B**) air relative humidity (%), (**C**) day light–sunshine (h) and (**D**) soil moisture (%) during the experiment.

**Figure 2 plants-09-01032-f002:**
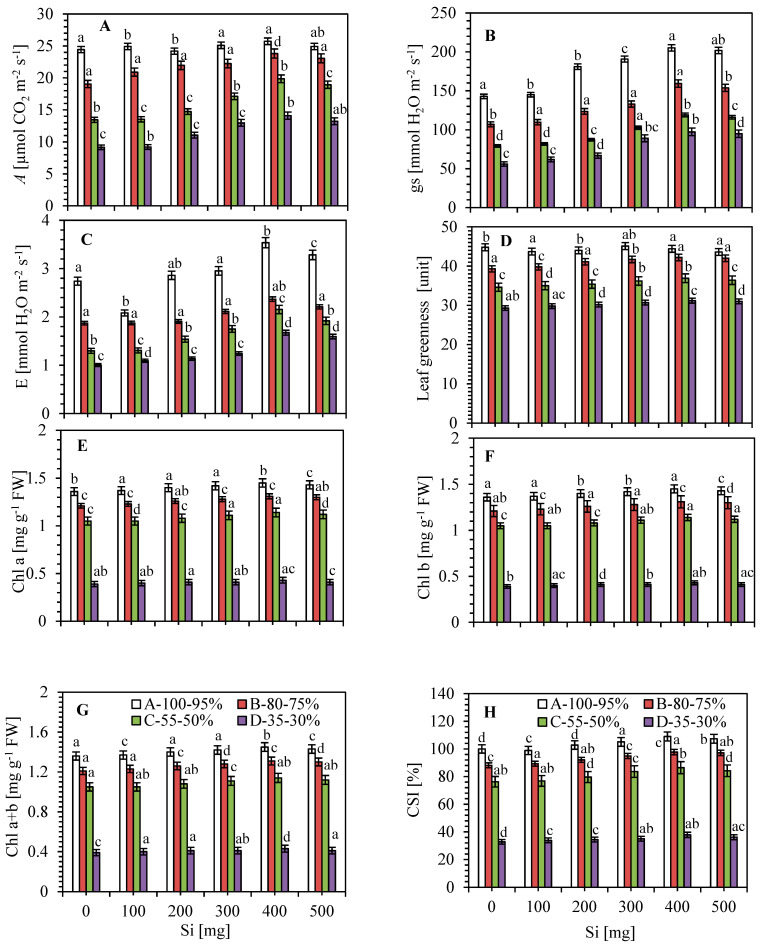
Changes of (**A**) net photosynthetic CO_2_ assimilation rate (*A*; µmol CO_2_ m^−1^s^−1^), (**B**) stomatal conductance (*gs*; mmol H_2_O m^−2^s^−1^), (**C**) transpiration rate (*E*; mmol H_2_O m^−2^s^−1^), (**D**) leaf greenness—SPAD value (units), (**E**–**G**) photosynthetic pigments (Chl; mg g^−1^ FW) and (**H**) chlorophyll stability index (CSI; %) after silicon application (0, 100, 200, 300, 400 and 500 mg L^−1^) against normal (100–95%) and limited water irrigations such as 80–75, 55–50 and 35–30% of soil moisture in sugarcane plants. Bar indicates ± SD (*n* = 5). Different letters represent significant differences at *p* < 0.05 level.

**Figure 3 plants-09-01032-f003:**
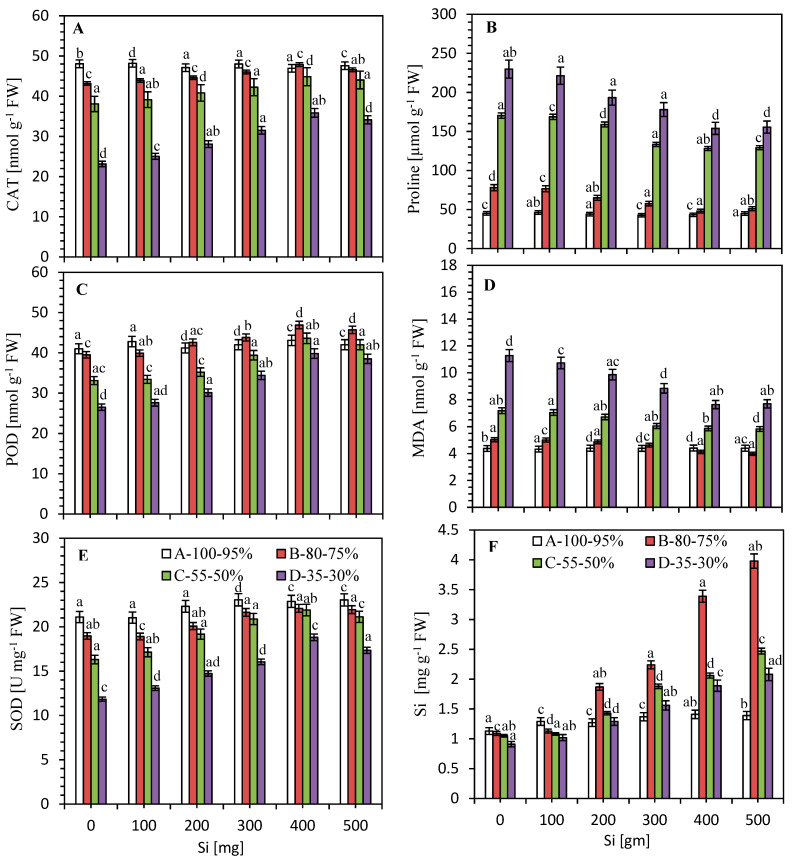
Effect of antioxidant enzymatic activities such as (**A**) catalase (CAT; nmol g^−1^ FW), (**C**) peroxidase (POD; nmol g^−1^ FW), (**E**) superoxide dismutase (SOD; unit mg^−1^ FW) and non-enzymatic activities, i.e., (**B**) proline content (µmol g^−1^ FW), (**D**) malondialdehyde content (MDA; nmol g^−1^ FW), and (**F**) silicon content (Si; mg g^−1^ FW) in sugarcane leaves on day 60 after normal and limited irrigation with silicon application (0, 100, 200, 300, 400 and 500 mg L^−1^). Vertical bars represent ± standard errors (*n* = 5).

**Table 1 plants-09-01032-t001:** Effect of silicon (0, 100, 200, 300, 400 and 500 mg Si L^−1^) on plant length (PL), Leaf number (LN), leaf area-expansion (LAE), and leaf water content (LWC) in sugarcane plants exposed to 100–95, 80–75, 55–50 and 35–30% of soil moisture (SM). Data are means ± SE (*n* = 5).

Silicon [mg L^−1^]	SML [%]	PL [cm]	LN	LAE [m^2^ plant^−1^]	LWC [%]
0	100–95	112 ± 10af	18 ± 3ac	0.7504 ± 0.006aj	92.85 ± 2.8ah
100		117 ± 13e	16 ± 2ej	0.7622 ± 0.003h	93.03 ± 1.9g
200		114 ± 8d	19 ± 2aeg	0.7501 ± 0.007al	93.08 ± 3.6aef
300		122 ± 5ij	18 ± 3be	0.7553 ± 0.008ad	93.01 ± 2.5ade
400		120 ± 9hi	18 ± 2ad	0.7789 ± 0.010cl	92.98 ± 2.0ah
500		118 ± 11akj	16 ± 3ai	0.7698 ± 0.008n	93.03 ± 3.1adi
0	80–75	83 ± 5adp	11 ± 1d	0.7019 ± 0.011ap	76.03 ± 2.7fn
100		85 ± 7al	11 ± 2eh	0.7141 ± 0.004o	76.45 ± 1.7ac
200		89 ± 11hi	13 ± 2fi	0.7188 ± 0.014ak	77.61 ± 2.0hi
300		95 ± 10aok	14 ± 3ghi	0.7250 ± 0.009ap	79.33 ± 2.3adg
400		101 ± 6am	14 ± 2d	0.7318 ± 0.006egi	79.87 ± 2.7ac
500		100 ± 8aeh	13 ± 2f	0.7276 ± 0.011gh	78.92 ± 1.8e
0	55–50	48 ± 3ef	8 ± 1ae	0.5009 ± 0.016d	63.01 ± 1.1ai
100		48 ± 5ad	8 ± 3i	0.5091 ± 0.019ef	64.11 ± 1.3n
200		50 ± 5an	9 ± 2aj	0.5108 ± 0.009ae	64.06 ± 0.9f
300		52 ± 7f	9 ± 1af	0.5209 ± 0.011mn	66.29 ± 1.0de
400		53 ± 4ab	11 ± 2n	0.5301 ± 0.013an	69.87 ± 1.3pc
500		53 ± 3a	10 ± 2ade	0.5248 ± 0.009ci	69.02 ± 0.8m
0	35–30	37 ± 2ce	6 ± 1aj	0.3678 ± 0.007fg	40.11 ± 0.5gh
100		37 ± 4l	6 ± 2cj	0.3706 ± 0.005hi	40.83 ± 0.8op
200		37 ± 4ki	6 ± 1n	0.3817 ± 0.007ae	41.51 ± 1.1ahi
300		40 ± 3aj	7 ± 2l	0.3873 ± 0.009afi	41.96 ± 0.7l
400		42 ± 2h	8 ± 2ap	0.4006 ± 0.005aj	43.07 ± 1.3aj
500		42 ± 3eg	8 ± 1ace	0.3903 ± 0.013ade	42.63 ± 1.2ace
Variation factor					
Si		**	NS	**	**
SML		NS	**	NS	NS
Si × SML		**	**	**	**

Different values within a column followed by different letters indicate significant (*p* < 0.05, Student t-test) differences. NS and ** represent non-significant and significant, respectively.

**Table 2 plants-09-01032-t002:** Quality of the sugarcane harvested at 60 days after Si application against limited water irrigation such as 100–95, 80–75, 55–50 and 35–30% of soil moisture. Lfm—leaf fresh, Ldm—leaf dry mass, Sfm—stem fresh, Sdm—stem dry mass, Rfm root fresh, Rdm—root dry mass, Tfm—total fresh, and Tdm—total dry mass. Data are means ± SE of three replicates in each irrigation levels.

Silicon [mg L^−1^ ]	SML [%]	Biomass [g]							
		Lfm	Ldm	Sfm	Sdm	Rfm	Rdm	Tfm	Tdm
0	100–95	212.55 ± 10.1ac	69.96 ± 3.3la	969.71 ± 9.3ad	229.78 ± 13.1ai	129.79 ± 4.1bc	51.11 ± 1.8bce	1312.05 ± 22.5gh	350.85 ± 5.8ae
100		210.85 ± 7.3pi	68.11 ± 2.3j	979.45 ± 7.1p	230.19 ± 9.8cd	132.59 ± 3.7ef	51.33 ± 1.1ci	1322.89 ± 19.1aei	349.63 ± 6.3bgi
200		215.25 ± 13.3od	70.18 ± 2.1kl	974.04 ± 7.7ci	230.08 ± 7.6bde	134.11 ± 3.3cd	52.08 ± 1.4de	1323.4 ± 17.8hi	352.34 ± 9.1ac
300		213.71 ± 6.9mn	68.98 ± 1.8ac	978.87 ± 6.8op	231.19 ± 10.1ae	137.03 ± 3.9a	53.21 ± 0.9e	1329.61 ± 23.3ef	353.38 ± 5.8de
400		218.13 ± 10.1ca	71.05 ± 1.3ac	979.89 ± 5.3ai	231.33 ± 12.5f	138.89 ± 2.8ac	53.95 ± 1.1ad	1336.91 ± 17.5dei	356.33 ± 7.4e
500		216.01 ± 7.9df	69.59 ± 1.9b	986.66 ± 8.1m	232.06 ± 11.3ad	141.07 ± 3.5ec	95.24 ± 1.7g	1343.74 ± 21.2e	396.89 ± 10.1bc
0	80–75	191.91 ± 3.9ad	61.85 ± 1.2fi	765.95 ± 5.8no	183.08 ± 7.6i	122.37 ± 4.1aei	43.08 ± 0.7af	1080.23 ± 13.1aef	288.01 ± 8.4ai
100		192.05 ± 4.2bl	62.03 ± 1.3ce	766.03 ± 6.3ad	183.23 ± 7.1bc	123.98 ± 2.2g	43.70 ± 0.9ig	1082.06 ± 9.8bc	288.96 ± 7.9ef
200		195.81 ± 4.1ai	63.81 ± 1.7g	769.07 ± 4.8l	183.87 ± 8.5g	124.08 ± 2.9j	44.01 ± 1.3kn	1088.96 ± 17.2ae	291.69 ± 4.7ak
300		196.64 ± 5.6ace	84.54 ± 2.3h	774.25 ± 5.1ai	184.11 ± 6.2ab	125.81 ± 3.1l	45.49 ± 1.1ah	1096.7 ± 10.2k	314.14 ± 5.8ajk
400		201.03 ± 4.1cn	85.76 ± 1.6bc	778.01 ± 4.9k	185.23 ± 5.3h	128.13 ± 2.2ah	46.33 ± 1.2al	1107.17 ± 17.8lm	317.32 ± 6.1agh
500		198.63 ± 3.9op	85.03 ± 2.1d	775.81 ± 6.3b	184.82 ± 7.1k	127.44 ± 1.8acn	46.08 ± 1.7an	1101.88 ± 13.3ao	315.93 ± 4.1al
0	55–50	112.55 ± 3.2ai	39.96 ± 2.3ef	543.30 ± 4.1c	113.80 ± 4.3lm	98.21 ± 1.1m	30.70 ± 0.9jk	754.06 ± 11.8an	184.46 ± 4.3aeh
100		112.85 ± 2.9ai	40.01 ± 1.1ac	544.08 ± 3.9ef	113.95 ± 3.6j	98.46 ± 1.9ajk	31.11 ± 0.3o	755.39 ± 13.3jl	185.07 ± 5.1ai
200		115.31 ± 2.6gh	40.64 ± 1.7fh	545.89 ± 5.5d	114.25 ± 2.8ao	99.03 ± 1.6ae	31.68 ± 0.9am	760.23 ± 9.1ai	186.57 ± 6.1aop
300		118.76 ± 3.3ah	41.18 ± 1.3ae	546.81 ± 2.8fh	114.53 ± 4.1acn	101.13 ± 1.5ao	33.03 ± 0.5ai	766.7 ± 13.5hik	188.74 ± 3.8n
400		123.96 ± 2.1k	41.91 ± 1.4a	551.11 ± 3.7h	115.08 ± 3.9p	104.28 ± 2.1ci	34.23 ± 0.5aeg	779.35 ± 17.4am	191.22 ± 5.2ace
500		120.72 ± 2.7j	40.88 ± 1.2m	548.25 ± 3.2aj	114.68 ± 5.7ace	102.34 ± 1.3ab	34.08 ± 0.4p	771.31 ± 9.6ae	189.64 ± 4.1am
0	35–30	55.85 ± 1.8ai	18.16 ± 0.8ap	413.15 ± 5.8g	83.67 ± 2.1ai	68.05 ± 1.1ai	15.95 ± 0.3ai	537.05 ± 5.8eg	117.78 ± 3.7ai
100		56.36 ± 2.3bc	18.21 ± 0.3lm	413.65 ± 4.2abe	84.03 ± 4.7bc	68.63 ± 1.3ap	16.16 ± 0.2ab	538.64 ± 9.2p	118.4 ± 3.7fgh
200		57.82 ± 1.9ac	18.91 ± 0.5ai	415.58 ± 3.3i	83.11 ± 2.7af	69.11 ± 1.4ac	16.93 ± 0.2ce	542.51 ± 9.3ce	118.95 ± 3.1ap
300		58.70 ± 1.1de	19.02 ± 0.4m	416.ac41 ± 3.9	83.29 ± 1.9ac	69.80 ± 0.9cd	17.21 ± 0.3a	544.91 ± 9.1ab	119.52 ± 1.9ac
400		61.08 ± 1.1af	19.93 ± 0.4ad	419.73ai ± 3.9	84.02 ± 1.1ai	71.23 ± 0.9ci	18.16 ± 0.3ao	552.04 ± 7.2ef	122.11 ± 2.3ae
500		61.01 ± 1.9ae	19.34 ± 0.2aeg	417.50 ± 4.3ade	83.91 ± 0.9adg	71.08 ± 1.2aef	18.11 ± 0.2de	549.59 ± 5.6de	121.36 ± 1.7ace

Different letters represent statistically significant differences (*p* < 0.05).

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
