# Peer review of "Silicon Supply Improves Leaf Gas Exchange, Antioxidant Defense System and Growth in Saccharum officinarum Responsive to Water Limitation"

_plants, 2020, doi:10.3390/plants9081032_

Round 1

Reviewer 1 Report

The manuscript presents the results of the studies of the effect of different concentrations of silicon on the parameters of  growth, photosynthesis and the activity of antioxidant enzymes in sugarcane (Saccharum officinarum) plants growing in water deficit conditions. The research is interesting, but the results are not new. This is another example confirming the positive effect of silicon present in the environment on the growth of plants under some abiotic stresses, e.g. water deficiency. However, presented results do not evolve molecular mechanism of silicon action on studied physiological processes. Similar studies are recently published by Verma et al.(2019) Biomed J Sci & Tech Res 15(2) (DOI: 10.26717/BJSTR.2019.15.002685) and are cited by the authors.

In my opinion there are some aspects of the manuscript that the authors should considerate. The discussion should be modified and improved particularly fragments on proline level and the activity of antioxidant enzymes. The proline level is poorly discussed. The Fig3B shows that the proline level increases in the tissues during limited irrigation. However, the observed increase is lower in plants treated the same water deficit in the presence of Si compared with plants untreated with Si. This result is not discussed.  Discussion section of SOD, CAT and POD activities is inconsistent and should be improved.

Furthermore, it is worth to notice that sugarcane is a C4 plant. Did the authors take this into account when planning the experiments and analyzing and discussing the results? This is not mentioned anywhere in the manuscript but it is very important in my opinion.

Minor point: Figures 2 and 3 should be improved. Statistical analysis are not visible.

Author Response

      Responses of Hon’ble Reviewer # 1 comments/ suggestions

  • In my opinion there are some aspects of the manuscript that the authors should considerate. The discussion should be modified and improved particularly fragments on proline level and the activity of antioxidant enzymes. The proline level is poorly discussed. The Fig3B shows that the proline level increases in the tissues during limited irrigation. However, the observed increase is lower in plants treated the same water deficit in the presence of Si compared with plants untreated with Si. This result is not discussed.  Discussion section of SOD, CAT and POD activities is inconsistent and should be improved.

[As per kind suggestion regarding proline and antioxidant activities, discussion section modified/ revised and included supporting findings]

  • Furthermore, it is worth to notice that sugarcane is a C4 plant. Did the authors take this into account when planning the experiments and analyzing and discussing the results? This is not mentioned anywhere in the manuscript but it is very important in my opinion.

[Related review incorporated, as advised]

  • Minor point: Figures 2 and 3 should be improved. Statistical analysis are not visible.

[Statistical analysis improved in figures]

Reviewer 2 Report

The manuscirpt focused on the influence of silicon on sugarcane, which is interesting and could be used in practice. However, it needs improvement before being accepted for publication. 

  1. the Introduction is wordy, which does not show why the study is conducted.
  2. the language needs improving, excessive statements are not clear or not easy to understand, e.g. what is "Limited number of irrigations"? (line 38); the statements "can be burned to generation of renewable energy" (line 67) and "significantly significant differences "are not right.
  3. table 1 should not be put in the Material and method
  4. the significance of difference among treatements (shown by letters) in tables and figers are not clear
  5. the discussion is not sufficient in the manuscript, the authors should not only focus on the results gotten by them. To explain why silicon is effective under drought is necessary.

Author Response

Responses of Hon’ble Reviewer # 2 comments/ suggestions

  • the Introduction is wordy, which does not show why the study is conducted.

[As per suggestion, related information included in the Introduction section]

  • the language needs improving, excessive statements are not clear or not easy to understand, e.g. what is "Limited number of irrigations"? (line 38); the statements "can be burned to generation of renewable energy" (line 67) and "significantly significant differences "are not right.

[MS language improved and corrections incorporated]

  • table 1 should not be put in the Material and method

[Table 1 shifted in result section]

  • the significance of difference among treatments (shown by letters) in tables and figures are not clear

[Statistical analysis corrected]

  • the discussion is not sufficient in the manuscript, the authors should not only focus on the results gotten by them. To explain why silicon is effective under drought is necessary.

[Related information included, as suggested]

Round 2

Reviewer 1 Report

All of the reviewer comments have been taken into account by the authors.

Author Response

All of the reviewer comments have been taken into account by the authors.

[NA]

Reviewer 2 Report

There are still a few problems to solve:

(1) the significance demonstrated by lettes in figure 2 and 3 are not clear, the ANOVA is done for different Si contents? or it is done with single Si concent to compare the difference among different watering treatments? there are obiviously mistakes, please check carefully.

(2) the author should provide detailed information of sampling in figure legends, as it is difficult to know when the enzyme activity and some metabolite level were tested;

(3) the abstract focused mainly on the influence of water stress, I think it should be improved to emphasize the effect of Si application, which is the novelty of this work;

(4)English in writing is still to be improved to avoid prolixity.

Author Response

Responses of the Hon'ble Reviewer comments/ suggestions as stated below:

  • the significance demonstrated by letters in figure 2 and 3 are not clear, the ANOVA is done for different Si contents? or it is done with single Si concent to compare the difference among different watering treatments? there are obiviously mistakes, please check carefully.

[Increased visibility/ font size of the significant letters in Fig. 2 & 3 both. The ANOVA is done for different Si levels with compare to control of each level]

  • the author should provide detailed information of sampling in figure legends, as it is difficult to know when the enzyme activity and some metabolite level were tested;

[Correction incorporated, as advised]

  • the abstract focused mainly on the influence of water stress, I think it should be improved to emphasize the effect of Si application, which is the novelty of this work;

[Thank for nice comment and suggestion. Abstract improved, as advised]

  • English in writing is still to be improved to avoid prolixity.

         [Language improved, as per kind suggestion]